# The application of cell-free DNA methylation patterns in critical illnesses: A protocol paper

**Shih-Han Kao[1,2], K. Greer Stumpf[1,2], Hunter A. Gaudio[1,2], John C. Greenwood[3], Samuel S. Shin[1,3], Frances S. Shofer[3], David H. Jang[3]***

**1** Resuscitation Science Center, The Children's Hospital of Philadelphia, Philadelphia, Pennsylvania, United States of America, **2** Department of Anesthesiology and Critical Care Medicine, Children's Hospital of Philadelphia, Philadelphia, Pennsylvania, United States of America, **3** Department of Emergency Medicine, Perelman School of Medicine, University of Pennsylvania, Philadelphia, Pennsylvania, United States of America

* david.jang@pennmedicine.uphs.edu

## Abstract

Biomarkers in clinical medicine are typically employed to gauge severity of disease, prognosis and to monitor response to treatment. While various biomarkers have been employed in clinical medicine with variable performance characteristics, the use of cell-free DNA (cfDNA) have gained increased traction as a novel biomarker in a wide range of disease states such as cancer and trauma. While the quantification of cfDNA have been correlated with disease severity, the use of methylation pattens of cfDNA can be used to localize the site of injury that may have implications regarding prognosis and therapeutics. We propose a procedure using samples in a swine model of cardiac arrest where carbon monoxide is being used as a therapeutic to demonstrate our method and feasibility to obtain plasma cfDNA methylation patterns to help identify tissue origin with potential application in critical care medicine.

## Introduction

Biomarkers in clinical medicine are typically employed to gauge severity of disease, prognosis and to monitor response to treatment [1]. Some common biomarkers used in clinical medicine include lactate to assess tissue hypoperfusion in sepsis or the use of troponin for cardiac tissue injury. More specific biomarkers in the field of medical toxicology include the use of carboxyhemoglobin (COHb) in carbon monoxide (CO) poisoning or an acetylcholinesterase activity level to gauge severity of organophosphate poisoning [2,3]. However, despite the prevalent use of these known biomarkers, their performance characteristics have been shown not to be reliable. Blood-based biomarkers should be highly dynamic with their diagnostic performance varying on internal and external changes. One exciting biomarker that have gained increased traction is the use of cell-free DNA (cfDNA).

**Data availability statement:** All relevant data are within the manuscript and its Supporting Information files.

**Funding:** R01HL166592

**Competing interests:** The authors have declared that no competing interests exist

The main source of cfDNA is active secretion from cells and can also be released upon cell death. cfDNA are derived from either nuclear cfDNA (n-cfDNA) or mitochondrial cfDNA (mt-cfDNA) [4]. Early research with cfDNA was in the field of oncology to detect specific sequences or mutations to guide chemotherapy for patients as rapidly dividing cells often release cfDNA [5,6]. CfDNA has also served a role in the prognosis of disease progression and as a response to treatment in other diseases. There is also controversy that cfDNA may also act as damage-associated molecular patterns (DAMPs) that can lead to sterile inflammation. While inflammation can have positive benefits such as tissue repair in the acute phase, long-term inflammation can lead to neurogenerative diseases and autoimmune diseases [7]. While it is possible cfDNA may directly lead to inflammation, data from supporting studies conclude that cfDNA does not directly contribute to disease progression but more likely serves as a biomarker of cellular stress or cell death [8,9].

There are two distinct advantages unique to cfDNA that many current biomarkers cannot accomplish: (1) cfDNA quantification: The total copy number of cfDNA has shown promise as a biomarker to assess severity of disease and may also correlate with clinically relevant parameters such as organ dysfunction and mortality. In addition, both n-cfDNA and mt-cfDNA percentage can be obtained from the total cfDNA copy number that have been both individually explored as a biomarker [10,11]; (2) cfDNA tissue origin: An important limitation for many biomarkers is that their origin of release is non-specific and does not directly assess the organ that may be injured. Lactate, for example, often indicates general cellular dysfunction from poor perfusion or mitochondrial injury but a limitation is that it is not clear where the lactate specifically comes from. The objective of this protocol is to describe the potential utilization of cfDNA as a biomarker in our swine model using CO as a therapy in cardiac arrest as an example of our method.

## Materials and methods

### Animals and overall study design

This is an ongoing swine study to demonstrate our protocol, all procedures were approved by the Institutional Animal Care and Use Committee at the Children's Hospital of Philadelphia (CHOP) and performed in accordance with the National Institutes of Health Guide for the Care and Use of Laboratory Animals (Protocol 1443). As an ongoing study, we have completed four animals out of the ten animals. Yorkshire pigs (weight ~ 10 kg and approximately 1–2 months of chronological age), equal numbers of each sex, were used. All pigs underwent an entrance exam that included a baseline physical exam, fecal occult testing for parasites and assessment by veterinary staff and were required to acclimate for a minimum of one day prior to any scheduled experiments. All animals were premedicated with 20 mg/kg ketamine, followed by inhaled 5% isoflurane in 100% oxygen through a snout mask to allow intubation for all peri-operative procedures described below. The protocol described in this peer-reviewed article is published on protocols.io, https://dx.doi.org/10.17504/protocols.io.5qpvo9p4bv4o/v1 and is included for printing as S1 File with this article.

## Perioperative procedures and monitoring

Ventilator settings consisted of a tidal volume 10–11mL/kg, positive end-expiratory pressure 5 cm $H_2O$ and respiratory rate titrated to achieve end-tidal $CO_2$ 38–42 mmHg. The right femoral artery and vein were cannulated for arterial pressure monitoring and central venous access. Isoflurane was weaned to 0.5–1% and sedation was maintained with the use of fentanyl (5 µg/kg/h) and dexmedetomidine (2 µg/kg/h) for the duration of the experiment. A 48 cm percutaneous tunneled catheter was placed into the right internal jugular vein to allow sampling of blood and administration of intravenous medication during the survivor period. All data were recorded with PowerLab 16/35 LabChart 8 Pro software from ADInstruments (Sydney, Australia).

## Cardiac arrest protocol

Our cardiac arrest protocol consisted of 8 min of untreated ventricular fibrillation (VF) followed by standardized Advanced Cardiac Life Support (ACLS) consisting of cardiopulmonary resuscitation (CPR) with first defibrillation taking place 2 min after CPR was initiated (10 min after the start of the VF arrest) every two min until return of spontaneous circulation (ROSC) or until 20 min of ACLS. Animals that achieved ROSC was maintained under general anesthesia to a $PaO_2$ 60–100 mmHg and a $PaCO_2$ 35–45 mmHg with predefined hemodynamic targets with IV fluids to achieve adequate intravascular volume status and norepinephrine to achieve target mean arterial pressure (MAP). Normothermia and continuous hemodynamic monitoring was maintained throughout the post resuscitation experimental period. If no ROSC was achieved, resuscitation was continued for an additional 10 min for a total of 20 min of CPR. If ROSC was achieved, CO treatment with 200 ppm was administrated for 2 hr for a total of 3 hr post-ROSC based on our previous protocol publication [12].

## Carbon monoxide treatment protocol

The assigned CO dose of 200 ppm was administered with a CO tank (244 cf) at 0–10L/min using a regulator with flow meter from Airgas (Radnor Township, PA, USA). Medical air was administered for controls. The CO concentration entering the endotracheal tube was monitored using an Inspector CO detector with a 0–2000 ppm range (Sensorcon, New York, USA). Sedation was maintained with the use of fentanyl (5 µg/kg/h) and dexmedetomidine (2 µg/kg/h) during the entire experiment. Our previous prior work has utilized CO doses of 400 ppm and 2000 ppm [12].

## Blood collection, cell-free DNA extraction, and bisulfite conversion

To illustrate our protocol levering our established swine model of cardiac arrest with the use of CO as a therapy, blood samples were collected following a standard protocol using cfDNA/cfRNA Preservative Norgen tubes (Norgen Biotek, Canada). The following protocol was based on Norgen Biotek's recommendation with respect to centrifugation speed and temperature. A total of 20 ml of whole blood were placed into two Norgen tubes, each tube was then gently inverted 5 times to allow gentle mixture of the preservative and the blood samples followed immediately for processing. Samples were then centrifuged at 1,600 x g for 10 min at 4°C to obtain a total of 10 mL of plasma on the upper layer. For cfDNA, the plasma was then centrifuged at 16,000 x g at RT to remove cell debris. The supernatant was then stored at -80°C before cfDNA extraction. cfDNA extraction was then performed with the NextPrep-Mag cfDNA isolation kit (PerkinElmer, MA), following the manufacturer's protocol. CfDNA quantification was performed using Qubit 1x dsDNA HS assay kit with the Qubit 4 fluorometer (Thermo Fisher Scientific). Swine cortex (PG-212) and blood (PG-705) genomic DNA (gDNA) samples were acquired from Zyagen as reference genomes. Samples were bisulfite converted using EZ DNA methylation-lightning kit (Zymo Research) before sequencing and data processing [12,13].

Libraries for the WGBS were created from the bisulfite-converted samples using the NEBNext Enzymatic Methyl-seq Kit (New England Biolabs) according to the manufacturer's protocol. Libraries were quantified with the Qubit 1x dsDNA

high sensitivity assay kit and the Qubit 4 fluorometer. The WGBS was conducted on the NovaSeq 6000 platform (Illumina) producing 2 x 150 paired end reads at 30 x coverage. Sequencing reads were analyzed as follows: quality checked, trimming, and aligned to the pig reference genome assembly, respectively. Bismark methylation extraction was applied to extract the methylation call for CpGs in each sample [14].

**Differential methylation analysis.** We used the Seqmonk analytic workflow to identify DMRs among tissue types (https://www.bioinformatics.babraham.ac.uk/projects/seqmonk/). All DMRs were based on a genomic window size of 25 CpGs. Methylation values were quantitated by the "bisulphite methylation over features" pipeline in Seqmonk. The following criteria were applied for the selection of the following DMRs. Brain-specific DMRs were defined as those regions in which methylation in the cortex differed from methylation in blood by more than 50% (differential (diff) methylation >50%). Positive diff methylation indicates relative hypermethylation of the brain-specific DMR relative to blood, whereas negative diff methylation indicates relative hypomethylation, with an FDR < 0.00005.

**Statistical analyses.** The EdgeR package in Seqmonk was applied to DMR analysis. The Benjamini-Hochberg procedure was carried out to adjust p-values for multiple comparisons.

## Result

### Cell-free DNA

We found that cfDNA methylation patterns were distinctively clustered at 2h post CO treatment and at the end of the observation and that the cfDNA components varied between the two groups [Fig 1]. The blood tissue served as a background tissue as most of the cfDNA fragments in plasma are derived from the apoptotic hematopoietic cells in healthy subjects [15]. A group of hypermethylated brain DMRs (in red) and a group of hypomethylated brain DMRs (in blue) [Fig 2]. Then a comparison of the cfDNA methylome to the methylation atlas and we found a hypomethylated brain DMR (BD). The level of unmethylated BD, which is the proportion of BD that is not methylated, significantly increased at 2h post sham treatment and at the end of observation [Fig 3]. On the contrary, the levels of unmethylated BD were kept constant during and after CO treatment.

A.

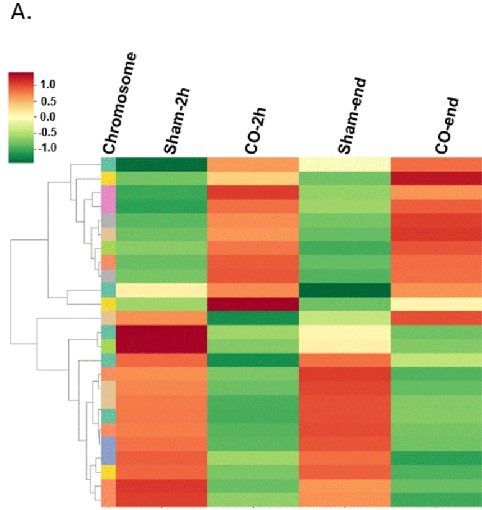

**Fig 1. Profiling of differentially methylation regions (DMRs) in sham and CO groups.** (A) Heatmap and dendrogram showing clustering of DMRs with similar methylation levels in samples: Sham-2h, CO-2h, Sham-end, CO-end. Sham-2h or CO-2h: hours of administering sham or CO treatment; sham-end or CO-end: termination of ROSC in sham or CO. *P < 0.05.

B.

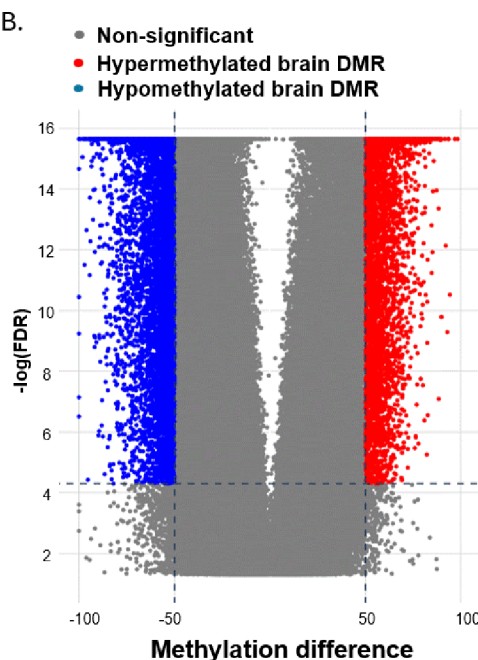

**Fig 2. Profiling of differentially methylation regions (DMRs) in sham and CO groups.** (B) Volcano plot for both brain hypermethylated (red) and hypomethylated (blue) DMRs in brain, compared with blood. *P < 0.05.

C.

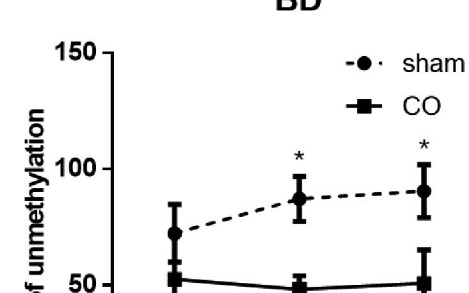

**Fig 3. Profiling of differentially methylation regions (DMRs) in sham and CO groups.** (C) Levels of unmethylated BD, a potential brain-specific biomarker, designated as BD, in the Sham and CO group. *P < 0.05.

## Discussion

To examine whether CO treatment makes a difference to animals suffering from cardiac arrest, we profiled the cfDNA methylome in Sham and CO groups using two animals in each group to illustrate our methods. To achieve this, we combined our expertise in cardiac arrest and CO poisoning as a therapy in cardiac arrest to illustrate the potential application

of cfDNA as an innovative biomarker. In our study we demonstrate the protocol and method to use cfDNA to help localize tissue source using our swine model of acute injury [12,13].

One of the strengths of cfDNA when compared to other biomarkers include the ability to use bulk concentrations of cfDNA to correlate with severity of disease illness similar to the use of lactate in sepsis [5]. For instance, the degree of lactate elevation can correlate with degree of tissue hypoperfusion in a wide range of cellular hypoxia. For example, the degree of lactate elevation can correlate with the severity of cyanide poisoning. Another example is the use of cholinesterase activity to gauge the severity of organophosphate (OP) poisoning [3,16]. As with many clinically used biomarkers, cholinesterase activity has poor correlation with severity. We compared the use of cfDNA against cholinesterase activity in a rodent model of chlorpyrifos and found that cfDNA was a more sensitive biomarker of OP poisoning [13].

Another advantage that cfDNA has to offer in addition to measuring cfDNA concentrations is also the ability to provide the tissue origin that may localize the site of injury. Others have leveraged the use of cfDNA to provide tissue localization information. One method is to measure the methylation pattern of cfDNA that may provide the tissue source of the DNA. Methylation of cytosine adjacent to guanine (CpG) is an essential component of cell-specific gene regulation and serves as a marker of cell identity. cfDNA molecules from loci carrying tissue-specific methylation can be used to identify cell death in a specific tissue [17]. Tissue-specific methylation on cfDNA can be used to identify brain cell death (from other sources such as blood cells) and to evaluate post-injury neuro-recovery in the brain and central nervous system in future studies.

CfDNA has been shown to be useful biomarkers in diseases such as traumatic brain injury and cancer [8,9,18]. Here, we investigated the possibility of utilizing cfDNA in monitoring other secondary organ injuries caused by cardiac arrest. We showcased a proof-of-concept brain-specific cfDNA biomarker in our data. We first found that the cfDNA methylation profiles were different between groups. We then established a brain methylation atlas and found that brain-specific cfDNA, BD, demonstrated differential methylation in response to intervention. This could potentially be correlated to clinical outcomes and serve as a useful biomarker in the future. We demonstrated that plasma cfDNA shows great potential to identify tissue-specific secondary injuries, and this feature makes cfDNA biomarkers appealing in the field of critical care, where most biomarkers still lack tissue specificity.

Since our interest was to illustrate cfDNA as a biomarker to gauge the secondary organ injury, particularly the brain damage, we built the swine brain methylation atlas using genomic DNA isolated from the swine brain and blood tissue. These data suggest that CO treatment might protect the brain from being further injured during cardiac arrest. In sum, brain-specific cfDNA can potentially serve as a biomarker to monitor the progression of the injury and the response to treatment. In conclusion, cfDNA biomarkers can provide critical insights into the effectiveness of therapeutic interventions, potentially guiding adjustments in clinical strategies to improve patient outcomes.

## Limitations

While this is primarily a protocol paper for application in other similar animal models of critical illnesses there are some limitations. While we highlighted the differences in the two groups looking at the DNA methylation patterns, the results may vary depending on the injury model. This protocol study provides proof of concept for the utility of cfDNA to study tissue involvement in critical illnesses that may help to guide management. Another limitation is now the time, and cost may be prohibitive in the acute phase of cardiac arrest but can have potential application in the post-arrest phase once the patient has been stabilized and prognostics tests are being performed. Finally, another current limitation is that we have methylation data from brain and blood, but not from other tissues. Therefore, we cannot determine whether the brain-specific DMRs are exclusive to the brain. It would be of great interest to collect other tissues and investigate secondary damage in cardiac arrest.

## Supporting information

**S1 File. The application of cell-free DNA methylation patterns in critical illnesses protocol.**
(PDF)



## Author contributions

**Conceptualization:** Shih-Han Kao, Hunter A. Gaudio, John C. Greenwood.

**Data curation:** Hunter A. Gaudio, David H. Jang, K. Greer Stumpf.

**Formal analysis:** John C. Greenwood, Frances S. Shofer, David H. Jang.

**Funding acquisition:** David H. Jang.

**Methodology:** Samuel S. Shin, K. Greer Stumpf.

**Resources:** David H. Jang.

**Supervision:** David H. Jang.

**Writing – original draft:** Shih-Han Kao, John C. Greenwood, Samuel S. Shin, Frances S. Shofer, David H. Jang.

**Writing – review & editing:** Shih-Han Kao, John C. Greenwood, Samuel S. Shin, Frances S. Shofer, David H. Jang.

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
