## [Decision Letter · Decision Letter 0]

19 Nov 2024

PONE-D-24-42835The application of cell-free DNA methylation patterns in critical illnesses protocolPLOS ONE

Dear Dr. Jang,

Thank you for submitting your manuscript to PLOS ONE. After careful consideration, we feel that it has merit but does not fully meet PLOS ONE’s publication criteria as it currently stands. Therefore, we invite you to submit a revised version of the manuscript that addresses the points raised during the review process.

When responding to reviewer comments the authors need to also improve the sampling procedure for the pigs, without forgetting typos/overall grammar improvement.==============================

We look forward to receiving your revised manuscript.

Kind regards,

Elingarami Sauli, PhD

Academic Editor

PLOS ONE

Journal Requirements: When submitting your revision, we need you to address these additional requirements. 1. Please ensure that your manuscript meets PLOS ONE's style requirements, including those for file naming. The PLOS ONE style templates can be found at https://journals.plos.org/plosone/s/file?id=wjVg/PLOSOne_formatting_sample_main_body.pdf and https://journals.plos.org/plosone/s/file?id=ba62/PLOSOne_formatting_sample_title_authors_affiliations.pdf 2. We note you have not yet provided a protocols.io PDF version of your protocol and/or a protocols.io DOI. When you submit your revision, please provide a PDF version of your protocol as generated by protocols.io (the file will have the protocols.io logo in the upper right corner of the first page) as a Supporting Information file. The filename should be S1_file.pdf, and you should enter “S1 File” into the Description field. Any additional protocols should be numbered S2, S3, and so on. Please also follow the instructions for Supporting Information captions [https://journals.plos.org/plosone/s/supporting-information#loc-captions]. The title in the caption should read: “Step-by-step protocol, also available on protocols.io.” Please assign your protocol a protocols.io DOI, if you have not already done so, and include the following line in the Materials and Methods section of your manuscript: “The protocol described in this peer-reviewed article is published on protocols.io (https://dx.doi.org/10.17504/protocols.io.[...]) and is included for printing purposes as S1 File.” You should also supply the DOI in the Protocols.io DOI field of the submission form when you submit your revision. If you have not yet uploaded your protocol to protocols.io, you are invited to use the platform’s protocol entry service [https://www.protocols.io/we-enter-protocols] for doing so, at no charge. Through this service, the team at protocols.io will enter your protocol for you and format it in a way that takes advantage of the platform’s features. When submitting your protocol to the protocol entry service please include the customer code PLOS2022 in the Note field and indicate that your protocol is associated with a PLOS ONE Lab Protocol Submission. You should also include the title and manuscript number of your PLOS ONE submission. 3. Please include a link to the protocols.io entry in your Methods section. 4. We noticed you have some minor occurrence of overlapping text with the following previous publication(s), which needs to be addressed: Cell-Free DNA as a Biomarker in a Rodent Model of Chlorpyrifos Poisoning Causing Mitochondrial Dysfunction - https://doi.org/10.1007/s13181-023-00956-0 (Among others) In your revision ensure you cite all your sources (including your own works), and quote or rephrase any duplicated text outside the methods section. Further consideration is dependent on these concerns being addressed. 5. Please note that PLOS ONE has specific guidelines on code sharing for submissions in which author-generated code underpins the findings in the manuscript. In these cases, we expect all author-generated code to be made available without restrictions upon publication of the work. Please review our guidelines at https://journals.plos.org/plosone/s/materials-and-software-sharing#loc-sharing-code and ensure that your code is shared in a way that follows best practice and facilitates reproducibility and reuse. 6. We note that the grant information you provided in the ‘Funding Information’ and ‘Financial Disclosure’ sections do not match.  When you resubmit, please ensure that you provide the correct grant numbers for the awards you received for your study in the ‘Funding Information’ section. 7. We note that your Data Availability Statement is currently as follows: All relevant data are within the manuscript and its Supporting Information files. Please confirm at this time whether or not your submission contains all raw data required to replicate the results of your study. Authors must share the “minimal data set” for their submission. PLOS defines the minimal data set to consist of the data required to replicate all study findings reported in the article, as well as related metadata and methods (https://journals.plos.org/plosone/s/data-availability#loc-minimal-data-set-definition). For example, authors should submit the following data: - The values behind the means, standard deviations and other measures reported;- The values used to build graphs;- The points extracted from images for analysis. Authors do not need to submit their entire data set if only a portion of the data was used in the reported study. If your submission does not contain these data, please either upload them as Supporting Information files or deposit them to a stable, public repository and provide us with the relevant URLs, DOIs, or accession numbers. For a list of recommended repositories, please see https://journals.plos.org/plosone/s/recommended-repositories. If there are ethical or legal restrictions on sharing a de-identified data set, please explain them in detail (e.g., data contain potentially sensitive information, data are owned by a third-party organization, etc.) and who has imposed them (e.g., an ethics committee). Please also provide contact information for a data access committee, ethics committee, or other institutional body to which data requests may be sent. If data are owned by a third party, please indicate how others may request data access. 8. Please include a separate caption for each figure in your manuscript. 9. Please include captions for your Supporting Information files at the end of your manuscript, and update any in-text citations to match accordingly. Please see our Supporting Information guidelines for more information: http://journals.plos.org/plosone/s/supporting-information. 10. Please review your reference list to ensure that it is complete and correct. If you have cited papers that have been retracted, please include the rationale for doing so in the manuscript text, or remove these references and replace them with relevant current references. Any changes to the reference list should be mentioned in the rebuttal letter that accompanies your revised manuscript. If you need to cite a retracted article, indicate the article’s retracted status in the References list and also include a citation and full reference for the retraction notice.

Reviewers' comments:

Reviewer's Responses to Questions

**Comments to the Author**

1. Does the manuscript report a protocol which is of utility to the research community and adds value to the published literature?

Reviewer #1: Yes

Reviewer #2: Yes

2. Has the protocol been described in sufficient detail?

To answer this question, please click the link to protocols.io in the Materials and Methods section of the manuscript (if a link has been provided) or consult the step-by-step protocol in the Supporting Information files.

The step-by-step protocol should contain sufficient detail for another researcher to be able to reproduce all experiments and analyses.

Reviewer #1: Yes

Reviewer #2: Yes

3. Does the protocol describe a validated method?

Reviewer #1: No

Reviewer #2: No

4. If the manuscript contains new data, have the authors made this data fully available?

Reviewer #1: No

Reviewer #2: No

**5. Is the article presented in an intelligible fashion and written in standard English?**

Reviewer #1: Yes

Reviewer #2: Yes

6. Review Comments to the Author

Reviewer #1: In this manuscript, the authors propose a procedure using samples in a swine model of cardiac arrest where carbon monoxide is being used as a therapeutic to demonstrate the method and feasibility to obtain plasma cfDNA methylation patterns to help identify tissue origin with potential application in critical care medicine. In general, the manuscript is well written, focused and proposes new role for liquid markers.

However, the authors should address some specific points:

1. The authors state that Yorkshire pigs (weight ~ 10 kg and 95 approximately 1-2 months of chronological age), equal numbers of each sex, were used. The number of animals and their allocation in groups/subgroups should be stated. Moreover, a short table showing how the animals were divided and in which groups would be advisable.

2. I could not access more extensive data that is a specific requirement of the journal. Please, provide these as an additional/complimentary data.

Reviewer #2: This manuscript by Kao et al. entitled “The application of cell-free DNA methylation patterns in critical illnesses protocol” aims to provide insights into the utility of cfDNA with accompanying methylome analysis as a tissue-specific biomarker in critical illness. Overall, this proof of concept was successfully achieved, however there are some clarifications that are required:

Major comments:

1) Please consider rephrasing your definition of Brain-specific DMRs. The definition becomes clear when viewing Fig B. However, when reviewing materials and methods, it suggests that a hypomethylated DMR is defined as a methylation difference of 0-49% in brain as compared to blood (page 16, line 163). This could be simplified to state “Brain-specific DMRs were defined as those regions in which methylation in the cortex differed from methylation in blood by more than 50% (differential (diff) methylation >50%). Positive diff methylation indicates relative hypermethylation of the brain-specific DMR relative to blood, whereas negative diff methylation indicates relative hypomethylation.”

2) The use of unmethylated BD (page 17, line 180) should be defined with further granularity. Does this refer to unmethylated regions (UMR) defined as 0-5% methylated which typically correspond to promoter-like elements (see citation below)? When referring to “levels of unmethylated BD” (page 17, line 182), is this referring to the proportion of BD that are unmethylated?

3) In addition to defining BD with increased granularity, Fig C would be further clarified by a Y-axis range of 0-100%. Additionally, the figure would be clearer if the Y-axis was titled “Percentage unmethylated BD” .

4) Has the raw data been made publicly available (i.e., deposited into a public repository)?

Minor comments:

1) Throughout the manuscript, please be cautious about use of plural vs singular (e.g., has vs have).

2) The conclusion on page 20, line 224, “…BD, can be used to trace the efficacy of CO treatment,” may be overstated as there are no clinical correlates provided in this study. It would potentially be more accurate to note that this study demonstrated the differential methylation of BD in response to intervention. This could potentially be correlated to clinical outcomes and serve as a useful biomarker in the future.

3) Though not necessary, it may be worthwhile to also highlight the short half-life of cfDNA which adds to its utility as a biomarker relative to other delayed biomarkers in clinical practice.

7. PLOS authors have the option to publish the peer review history of their article (what does this mean? ). If published, this will include your full peer review and any attached files.

**Do you want your identity to be public for this peer review?** For information about this choice, including consent withdrawal, please see our Privacy Policy .

Reviewer #1: No

Reviewer #2: No

---

## [Author Response · Author response to Decision Letter 1]

25 Dec 2024

PONE-D-24-42835

The application of cell-free DNA methylation patterns in critical illnesses protocol

PLOS ONE

Dear PLOS ONE,

Thank-you for the reviewer comments to improve this paper. We have submitted a revised manuscript that should address all of the feedback. Please let us know if you require any additional information and/or changes.

We have included the following for this submission:

· [ ] A rebuttal letter that responds to each point raised by the academic editor and reviewer(s). You should upload this letter as a separate file labeled 'Response to Reviewers'.

· [ ] A marked-up copy of your manuscript that highlights changes made to the original version. You should upload this as a separate file labeled 'Revised Manuscript with Track Changes'.

· [ ] An unmarked version of your revised paper without tracked changes. You should upload this as a separate file labeled 'Manuscript'.

For this submission we also have addressed the following and/or respond to the following:

[x] We note you have not yet provided a protocols.io PDF version of your protocol and/or a protocols.io DOI. When you submit your revision, please provide a PDF version of your protocol as generated by protocols.io (the file will have the protocols.io logo in the upper right corner of the first page) as a Supporting Information file. The filename should be S1_file.pdf, and you should enter “S1 File” into the Description field. Any additional protocols should be numbered S2, S3, and so on. Please also follow the instructions for Supporting Information captions [https://journals.plos.org/plosone/s/supporting-information#loc-captions]. The title in the caption should read: “Step-by-step protocol, also available on protocols.io.”

Please assign your protocol a protocols.io DOI, if you have not already done so, and include the following line in the Materials and Methods section of your manuscript: “The protocol described in this peer-reviewed article is published on protocols.io (https://dx.doi.org/10.17504/protocols.io.[...]) and is included for printing purposes as S1 File.” You should also supply the DOI in the Protocols.io DOI field of the submission form when you submit your revision.

If you have not yet uploaded your protocol to protocols.io, you are invited to use the platform’s protocol entry service [https://www.protocols.io/we-enter-protocols] for doing so, at no charge. Through this service, the team at protocols.io will enter your protocol for you and format it in a way that takes advantage of the platform’s features. When submitting your protocol to the protocol entry service please include the customer code PLOS2022 in the Note field and indicate that your protocol is associated with a PLOS ONE Lab Protocol Submission. You should also include the title and manuscript number of your PLOS ONE submission.

RESPONSE: Statistical analyses. The EdgeR package in Seqmonk was applied to DMR analysis. The Benjamini-Hochberg procedure was carried out to adjust p-values for multiple comparisons The protocol described in this peer-reviewed article is published on protocols.io.: (https://www.protocols.io/view/the-application-of-cell-free-dna-methylation-patte-dwau7aew)

[x] Please include a link to the protocols.io entry in your Methods section.

RESPONSE:

[x] We noticed you have some minor occurrence of overlapping text with the following previous publication(s), which needs to be addressed:

Cell-Free DNA as a Biomarker in a Rodent Model of Chlorpyrifos Poisoning Causing Mitochondrial Dysfunction - https://doi.org/10.1007/s13181-023-00956-0

In your revision ensure you cite all your sources (including your own works), and quote or rephrase any duplicated text outside the methods section. Further consideration is dependent on these concerns being addressed. In addition, the methods of this paper

RESPONSE: We revised the introduction section to ensure there is no overlap with our cited publication listed above. Also, for the methods we made minor revisions more specific to in swine model as opposed to the use of cfDNA in rodents which is what the paper above describes. The methods from the paper below is focused in the use in our swine model and DNA methylation pattern which is not present in our previous publication cited above.

To illustrate our protocol levering our established swine model of cardiac arrest with the use of CO as a therapy, blood samples were then collected following a standard protocol using cfDNA/cfRNA Preservative Norgen tubes (Norgen biotek, Canada). Briefly, a total of 20 ml of whole blood placement into the Norgen tubes, each tube was then gently inverted 5 times to allow gentle mixture of the preservative and the blood samples. Samples were then centrifuged at 1,600 x g for 10 min at 4°C to obtain plasma on the upper layer. For cfDNA, the plasma was then centrifuged at 16,000 x g at RT to remove cell debris. The supernatant was then stored at -80°C before cfDNA extraction. cfDNA extraction was then performed with the NextPrep-Mag cfDNA isolation kit (PerkinElmer, MA), following the manufacturer’s protocol. CfDNA quantification was performed using Qubit 1x dsDNA HS assay kit with the Qubit 4 fluorometer (Thermo Fisher Scientific). Swine cortex (PG-212) and blood (PG-705) genomic DNA (gDNA) samples were acquired from Zyagen as reference genomes. Samples were bisulfite converted using EZ DNA methylation-lightning kit (Zymo Research) before sequencing and data processing.12,13

Libraries for the WGBS were created from the bisulfite-converted samples using the NEBNext Enzymatic Methyl-seq Kit (New England Biolabs) according to the manufacturer’s protocol. Libraries were quantified with the Qubit 1x dsDNA high sensitivity assay kit and the Qubit 4 fluorometer. The WGBS was conducted on the NovaSeq 6000 platform (Illumina) producing 2 x 150 paired end reads at 30 x coverage. Sequencing reads were analyzed as follows: quality checked, trimmming, and aligned to the pig reference genome assembly, respectively. Bismark methylation extraction was applied to extract the methylation call for CpGs in each sample.14

Differential Methylation Analysis. We used the Seqmonk analytic workflow to identify DMRs among tissue types (https://www.bioinformatics.babraham.ac.uk/projects/seqmonk/). All DMRs were based on a genomic window size of 25 CpGs. Methylation values were quantitated by the “bisulphite methylation over features” pipeline in Seqmonk. The following criteria were applied for the selection of the following DMRs. Brain-specific hypermethylated DMRs were defined when the amount of methylation in cortex exceeded that of blood by more than 50 (differential (diff) methylation >50), and hypomethylated DMRs were defined as diff methylation < 50, with an FDR < 0.00005.

Statistical analyses. The EdgeR package in Seqmonk was applied to DMR analysis. The Benjamini-Hochberg procedure was carried out to adjust p-values for multiple comparisons.

[x] Please note that PLOS ONE has specific guidelines on code sharing for submissions in which author-generated code underpins the findings in the manuscript. In these cases, we expect all author-generated code to be made available without restrictions upon publication of the work. Please review our guidelines at https://journals.plos.org/plosone/s/materials-and-software-sharing#loc-sharing-code and ensure that your code is shared in a way that follows best practice and facilitates reproducibility and reuse.

RESPONSE: We did not have any code involved with this study did upload our raw data that is publicly available on Mendeley Data under the same title of this manuscript.

[x] We note that the grant information you provided in the ‘Funding Information’ and ‘Financial Disclosure’ sections do not match.

RESPONSE: I put in my funding (R01HL166592) under both Financial Disclosure and Funding information.

[x] We note that your Data Availability Statement is currently as follows: All relevant data are within the manuscript and its Supporting Information files.

RESPONSE: All our raw data is on Mendeley Data under the same title of this manuscript that is publicly available.

[x] Please include a separate caption for each figure in your manuscript.

RESPONSE: We now have separate caption for each figure

[x] Please include captions for your Supporting Information files at the end of your manuscript, and update any in-text citations to match accordingly. Please see our Supporting Information guidelines for more information: http://journals.plos.org/plosone/s/supporting-information.

RESPONSE: This protocol paper does not contain and supporting information files.

[x] Please review your reference list to ensure that it is complete and correct. If you have cited papers that have been retracted, please include the rationale for doing so in the manuscript text, or remove these references and replace them with relevant current references. Any changes to the reference list should be mentioned in the rebuttal letter that accompanies your revised manuscript. If you need to cite a retracted article, indicate the article’s retracted status in the References list and also include a citation and full reference for the retraction notice.

RESPONSE: Our reference is complete and correct upon second review.

Response to reviewer comments:

Reviewer #1: In this manuscript, the authors propose a procedure using samples in a swine model of cardiac arrest where carbon monoxide is being used as a therapeutic to demonstrate the method and feasibility to obtain plasma cfDNA methylation patterns to help identify tissue origin with potential application in critical care medicine. In general, the manuscript is well written, focused and proposes new role for liquid markers.

However, the authors should address some specific points:

1. The authors state that Yorkshire pigs (weight ~ 10 kg and 95 approximately 1-2 months of chronological age), equal numbers of each sex, were used. The number of animals and their allocation in groups/subgroups should be stated. Moreover, a short table showing how the animals were divided and in which groups would be advisable.

RESPONSE: This was a protocol manuscript to described for a future upcoming study and we performed on two animals for each group to highlight our methods. We revised the results section to include this information to highlight this fact

Result section under cell-free DNA section:

To examine whether CO treatment makes a difference to animals suffering from cardiac arrest, we profiled the cfDNA methylome in Sham and CO groups using two animals in each group to illustrate our methods

2. I could not access more extensive data that is a specific requirement of the journal. Please, provide these as an additional/complimentary data.

RESPONSE: All our raw data is now publicly available on Mendeley Data under the same title of this manuscript

Reviewer #2: This manuscript by Kao et al. entitled “The application of cell-free DNA methylation patterns in critical illnesses protocol” aims to provide insights into the utility of cfDNA with accompanying methylome analysis as a tissue-specific biomarker in critical illness. Overall, this proof of concept was successfully achieved, however there are some clarifications that are required:

Major comments:

1) Please consider rephrasing your definition of Brain-specific DMRs. The definition becomes clear when viewing Fig B. However, when reviewing materials and methods, it suggests that a hypomethylated DMR is defined as a methylation difference of 0-49% in brain as compared to blood (page 16, line 163). This could be simplified to state “Brain-specific DMRs were defined as those regions in which methylation in the cortex differed from methylation in blood by more than 50% (differential (diff) methylation >50%). Positive diff methylation indicates relative hypermethylation of the brain-specific DMR relative to blood, whereas negative diff methylation indicates relative hypomethylation.”

RESPONSE: We revised the Diferentual Methylation Analysis section as recommended above to address the reviewer comments.

Methods under Differential Methylation Analysis section

We used the Seqmonk analytic workflow to identify DMRs among tissue types (https://www.bioinformatics.babraham.ac.uk/projects/seqmonk/). All DMRs were based on a genomic window size of 25 CpGs. Methylation values were quantitated by the “bisulphite methylation over features” pipeline in Seqmonk. The following criteria were applied for the selection of the following DMRs. Brain-specific DMRs were defined as those regions in which methylation in the cortex differed from methylation in blood by more than 50% (differential (diff) methylation >50%). Positive diff methylation indicates relative hypermethylation of the brain-specific DMR relative to blood, whereas negative diff methylation indicates relative hypomethylation, with an FDR < 0.00005.

2) The use of unmethylated BD (page 17, line 180) should be defined with further

granularity. Does this refer to unmethylated regions (UMR) defined as 0-5% methylated which typically correspond to promoter-like elements (see citation below)? When referring to “levels of unmethylated BD” (page 17, line 182), is this referring to the proportion of BD that are unmethylated?

RESPONSE: See previous comment and response where we revised that section to also revise with this comment as well.

3) In addition to defining BD with increased granularity, Fig C would be further clarified by a Y-axis range of 0-100%. Additionally, the figure would be clearer if the Y-axis was titled “Percentage unmethylated BD” .

RESPONSE: We revised based on the above comment and includes this updated figure.

4) Has the raw data been made publicly available (i.e., deposited into a public repository)?

RESPONSE: We uploaded the raw data for all three figures to Mendeley Data

Minor comments:

1) Throughout the manuscript, please be cautious about use of plural vs singular (e.g., has vs have).

RESPONSE: As seen in our manuscript we revised to increase clarity of our manuscript

2) The conclusion on page 20, line 224, “…BD, can be used to trace the efficacy of CO treatment,” may be overstated as there are no clinical correlates provided in this study. It would potentially be more accurate to note that this study demonstrated the differential methylation of BD in response to intervention.

---

## [Decision Letter · Decision Letter 1]

11 Feb 2025

PONE-D-24-42835R1The application of cell-free DNA methylation patterns in critical illnesses protocolPLOS ONE

Dear Dr. Jang,

Thank you for submitting your manuscript to PLOS ONE. After careful consideration, we feel that it has merit but does not fully meet PLOS ONE’s publication criteria as it currently stands. Therefore, we invite you to submit a revised version of the manuscript that addresses the points raised during the review process.

When responding to reviewer comments, the authors should improve the materials and methods sections, by properly detailing the methods, including following proper sample collection procedures. The authors should also improve the discussion of obtained results.

We look forward to receiving your revised manuscript.

Kind regards,

Elingarami Sauli, PhD

Academic Editor

PLOS ONE

Journal Requirements:

Reviewers' comments:

Reviewer's Responses to Questions

**Comments to the Author**

1. Does the manuscript report a protocol which is of utility to the research community and adds value to the published literature?

Reviewer #1: Yes

Reviewer #3: Yes

2. Has the protocol been described in sufficient detail?

To answer this question, please click the link to protocols.io in the Materials and Methods section of the manuscript (if a link has been provided) or consult the step-by-step protocol in the Supporting Information files.

The step-by-step protocol should contain sufficient detail for another researcher to be able to reproduce all experiments and analyses.

Reviewer #1: Yes

Reviewer #3: Partly

3. Does the protocol describe a validated method?

Reviewer #1: No

Reviewer #3: No

4. If the manuscript contains new data, have the authors made this data fully available?

Reviewer #1: Yes

Reviewer #3: No

**5. Is the article presented in an intelligible fashion and written in standard English?**

Reviewer #1: Yes

Reviewer #3: **No: ** The writing is often unclear and disorganized, which makes it seem unprofessional. The English needs to be improved as well.

6. Review Comments to the Author

Reviewer #1: The authors addressed all points raised, and modified the manuscript accordingly. The manuscript can now be published.

Reviewer #3: The manuscript delineates a protocol for the analysis of cell-free DNA methylome in critical care medicine. This method has been evaluated in an animal model and shows promise for translation to human applications.

However, the manuscript's language is often cumbersome, lacking a coherent structure, and gives an overall unprofessional impression.The English language needs to be refined, and it is strongly recommended that the text undergo a thorough review and correction by a scientific editor.

The title is misleading; it should be corrected to reflect the nature of the study.

Introduction:

The utilization of cfDNA for the purpose of tracking genetic alterations is not pertinent to the subject at hand.

The objective of the study is not adequately delineated.

Abbreviations must be clarified at the first occurrence, e.g., n-cfDNA, mtDNA, etc.

Materials and methods:

The term "ongoing large animal study" is employed, yet it is evident that a total of four animals were utilized in the study. This inaccuracy must be rectified.

The procedure for blood processing is inadequately detailed, and the time frame between sample collection and centrifugation, the duration of the centrifugation process, the rationale for utilizing RT instead of 4°C, and the quantity of plasma samples utilized for cfDNA extraction should be clarified.

Has it been tested that the "brain-specific" DMRs are not specific to other tissues (exlcuding blood)?

The Results section is inadequately structured and not sufficiently explained. Please share more details.

The discussion section is similarly deficient in clarity and depth. References should be added to support the statements made in the discussion section.

7. PLOS authors have the option to publish the peer review history of their article (what does this mean? ). If published, this will include your full peer review and any attached files.

**Do you want your identity to be public for this peer review?** For information about this choice, including consent withdrawal, please see our Privacy Policy .

Reviewer #1: No

Reviewer #3: No

---

## [Author Response · Author response to Decision Letter 2]

19 Feb 2025

PONE-D-24-42835

The application of cell-free DNA methylation patterns in critical illnesses protocol

PLOS ONE

Dear PLOS ONE,

Thank-you for the reviewer comments to improve this paper and request for minor revisions. We have submitted a second revised manuscript that should address all the feedback. Please let us know if you require any additional information and/or changes.

We have included the following for this submission:

· [X] A rebuttal letter that responds to each point raised by the academic editor and reviewer(s). You should upload this letter as a separate file labeled 'Response to Reviewers'.

· [X] A marked-up copy of your manuscript that highlights changes made to the original version. You should upload this as a separate file labeled 'Revised Manuscript with Track Changes'.

· [X] An unmarked version of your revised paper without tracked changes. You should upload this as a separate file labeled 'Manuscript'.

For this submission we also have addressed the following and/or respond to the following:

1. General Comment: When responding to reviewer comments, the authors should improve the materials and methods sections, by properly detailing the methods, including following proper sample collection procedures. The authors should also improve the discussion of obtained results.

RESPONSE: We have revised the manuscript to make the recommended changes largely in response to Reviewer 3 below. As can be seen in our tracked changed manuscript that focuses on the revision of material and methods and discussion. Please see below for the specific changes.

2. Reviewer #1: The authors addressed all points raised and modified the manuscript accordingly. The manuscript can now be published.

RESPONSE: Thank-you for the positive comments and that the manuscript can now be published.

3. Reviewer #3: The manuscript delineates a protocol for the analysis of cell-free DNA methylome in critical care medicine. This method has been evaluated in an animal model and shows promise for translation to human applications.

However, the manuscript's language is often cumbersome, lacking a coherent structure, and gives an overall unprofessional impression. The English language needs to be refined, and it is strongly recommended that the text undergo a thorough review and correction by a scientific editor.

Response: We have revised the manuscript and as seen in our tracked change manuscript we have ensured enhanced readability.

The title is misleading; it should be corrected to reflect the nature of the study.

Response: We have changed the title to “The application of cell-free DNA methylation patterns in critical illnesses: A protocol paper.“

This is in line with similar protocol publications with PLOS One such as:

1. An evaluation of the Live Well Erie workforce development program, jumping off the benefits cliff: A protocol paper.

2. Prediction of Ovarian Hyperstimulation Syndrome in Patients Treated with Corifollitropin alfa or rFSH in a GnRH Antagonist Protocol.

1. Introduction:

The utilization of cfDNA for the purpose of tracking genetic alterations is not pertinent to the subject at hand.

The objective of the study is not adequately delineated.

Response: Based on the above comment we removed “The short half-life of cfDNA makes it an appealing tool for tracking genetic changes in real time, adding to its utility as a biomarker relative to other delayed biomarkers in clinical practice.”

We also clearly stated the objective of the study as follows:

1. Introduction Section Last Paragraph

“The objective of this protocol study is to describe the potential utilization of cfDNA as a biomarker in our swine model using carbon monoxide as a therapy in cardiac arrest as an example of our method.”

2. Abbreviations must be clarified at the first occurrence, e.g., n-cfDNA, mtDNA, etc.

Response: We have revised to what is recommended above. For example:

Introduction Paragraph 2

“The main source of cfDNA is typically actively secreted by cells and can also be released by cells upon death. cfDNA are derived from either nuclear cfDNA (n-cfDNA) or mitochondrial cfDNA (mt-cfDNA). Early research with cfDNA was in the field of oncology to detect specific sequences or mutations to guide chemotherapy for patients as rapidly dividing cells often release cfDNA. CfDNA has also served a role in the prognosis of disease progression as well as response to treatment in other disease states. There is also controversy that cfDNA may also act as a damage-associated molecular patterns (DAMPs) that can lead to sterile inflammation. While inflammation can have positive benefits such as tissue repair in the acute phase, long-term inflammation can lead to neurogenerative diseases and autoimmune diseases. While it is possible cfDNA may directly lead to inflammation, data from supporting studies conclude that cfDNA does not directly contribute to disease progression but more likely serves as a biomarker of cellular stress or cell death.”

3. Materials and methods:

The term "ongoing large animal study" is employed, yet it is evident that a total of four animals were utilized in the study. This inaccuracy must be rectified.

Response: We clarified to state as an ongoing study we have completed 4 animals with 6 more animals that are planned and that we used the collected data to illustrate the methods in this paper.

Material and Methods

“In this ongoing swine study to demonstrate our protocol, all procedures were approved by the Institutional Animal Care and Use Committee at the Children’s Hospital of Philadelphia (CHOP) and performed in accordance with the National Institutes of Health Guide for the Care and Use of Laboratory Animals (Protocol 1443). As an ongoing study, we have completed four animals out of the ten animals.

4. The procedure for blood processing is inadequately detailed, and the time frame between sample collection and centrifugation, the duration of the centrifugation process, the rationale for utilizing RT instead of 4°C, and the quantity of plasma samples utilized for cfDNA extraction should be clarified.

Response: We revised the relevant section to include the recommendations above

Blood collection, cell-free DNA extraction, and bisulfite conversion

To illustrate our protocol levering our established swine model of cardiac arrest with the use of CO as a therapy, blood samples were then collected following a standard protocol using cfDNA/cfRNA Preservative Norgen tubes (Norgen Biotek, Canada). The following protocol is based on Norgen Biotek’s recommendation with respect to centrifugation speed and temperature. A total of 20 ml of whole blood placement into two Norgen tubes, each tube was then gently inverted 5 times to allow gentle mixture of the preservative and the blood samples followed immediately for processing. Samples were then centrifuged at 1,600 x g for 10 min at 4°C to obtain a total of 10 mL of plasma on the upper layer. For cfDNA, the plasma was then centrifuged at 16,000 x g at RT to remove cell debris. The supernatant was then stored at -80°C before cfDNA extraction. cfDNA extraction was then performed with the NextPrep-Mag cfDNA isolation kit (PerkinElmer, MA), following the manufacturer’s protocol. CfDNA quantification was performed using Qubit 1x dsDNA HS assay kit with the Qubit 4 fluorometer (Thermo Fisher Scientific). Swine cortex (PG-212) and blood (PG-705) genomic DNA (gDNA) samples were acquired from Zyagen as reference genomes. Samples were bisulfite converted using EZ DNA methylation-lightning kit (Zymo Research) before sequencing and data processing.

5. Has it been tested that the "brain-specific" DMRs are not specific to other tissues (excluding blood)?

Response: We appreciate the reviewer’s comment. So far, we have methylation data from brain and blood, but not from other tissues. Therefore, we cannot determine whether the brain-specific DMRs are exclusive to the brain. It would be of great interest to collect other tissues and investigate secondary damage in cardiac arrest.

6. The Results section is inadequately structured and not sufficiently explained. Please share more details.

Response: We revised the results section with the above feedback

Result

We found that cfDNA methylation patterns were distinctively clustered at 2h post CO treatment and at the end of the observation andthat the cfDNA components varied between the two groups [Fig A].. The blood tissue served as a background tissue as most of the cfDNA fragments in plasma are derived from the apoptotic hematopoietic cells in healthy subjects. A group of hypermethylated brain DMRs (in red) and a group of hypomethylated brain DMRs (in blue) [Fig B]. Then a comparison of the cfDNA methylome to the methylation atlas and we found a hypomethylated brain DMR (BD). The level of unmethylated BD, which is the proportion of BD that is not methylated, significantly increased at 2h post sham treatment and at the end of observation [Fig. C]. On the contrary, the levels of unmethylated BD were kept constant during and after CO treatment

7. The discussion section is similarly deficient in clarity and depth. References should be added to support the statements made in the discussion section.

Response: We clarified and revised the entire discussion with the appropriate references.

Discussion

To examine whether CO treatment makes a difference to animals suffering from cardiac arrest, we profiled the cfDNA methylome in Sham and CO groups using two animals in each group to illustrate our methods.

To achieve this, we combined our expertise in cardiac arrest and CO poisoning as a therapy in cardiac arrest to primarily illustrate the potential application of cfDNA as an innovative biomarker. In our study we demonstrate the protocol and method to use cfDNA to help localize tissue source using our swine model of acute injury.12,13

One of the strengths of cfDNA when compared to other biomarkers include the ability to use bulk concentrations of cfDNA to correlate with severity of disease illness such as the use of lactate in sepsis.5 For instance, the degree of lactate elevation can correlate with degree of tissue hypoperfusion in a wide range of cellular hypoxia. For example, the degree of lactate elevation can correlate with the degree of cyanide poisoning. Another example where biomarker levels are used is the use of cholinesterase activity to gauge the severity of organophosphate (OP) poisoning.3,16 As with many clinically used biomarkers, cholinesterase activity has poor correlation with severity. We compared the use of cfDNA against cholinesterase activity in a rodent model of chlorpyrifos and found that cfDNA was a more sensitive biomarker of OP poisoning.13

Another advantage that cfDNA has to offer in addition to measuring cfDNA concentrations is also the ability to provide the tissue origin that may localize the site of injury. Others have leveraged the use of cfDNA to provide tissue localization information. One method that others have investigated is to measure the methylation pattern of cfDNA that may provide the tissue source of the DNA. Methylation of cytosine adjacent to guanine (CpG) is an essential component of cell-specific gene regulation and serves as a marker of cell identity. cfDNA molecules from loci carrying tissue-specific methylation can be used to identify cell death in a specific tissue.17 Tissue-specific methylation on cfDNA can be potentially used to identify brain cell death (from other sources such as blood cells) and to evaluate post-injury neuro-recovery in the brain and central nervous system in future studies.

CfDNA has been shown to be useful biomarkers in diseases such as traumatic brain injury and cancer.8,9,18 Here, we want to explore the possibility of utilizing cfDNA in monitoring other secondary organ injuries caused by cardiac arrest. We showcased a proof-of-concept brain-specific cfDNA biomarker in our data. We first found that the cfDNA methylation profiles were different between groups. We then established a brain methylation atlas and found that brain-specific cfDNA, BD, demonstrated differential methylation in response to intervention. This could potentially be correlated to clinical outcomes and serve as a useful biomarker in the future. We demonstrated that plasma cfDNA shows great potential to identify tissue-specific secondary injuries, and this feature makes cfDNA biomarkers appealing in the field of critical care, where most biomarkers still lack tissue specificity.

Since our interest was to illustrate cfDNA as a biomarker to gauge the secondary organ injury, particularly the brain damage, we built the swine brain methylation atlas using genomic DNA isolated from the swine brain and blood tissue. These data suggest that CO treatment might protect the brain from being further injured during cardiac arrest. In sum, brain-specific cfDNA can potentially serve as a biomarker to monitor the progression of the injury and the response to treatment. In conclusion, cfDNA biomarkers can provide critical insights into the effectiveness of therapeutic interventions, potentially guiding adjustments in clinical strategies to improve patient outcomes.

Limitations

While this is primarily a protocol paper for application in other similar animal models of critical illnesses there are some limitations to be aware of. While we highlighted the differences in the two groups looking at the DNA methylation patterns, the results may vary depending on the injury model. This protocol study provides proof of concept for the utility of cfDNA to study tissue involvement in critical illnesses that may help to guide management. Another limitation is now the time, and cost may be prohibitive in the acute phase of cardiac arrest but can have potential application in the post-arrest phase once the patient has been stabilized and prognostics tests are being performed. Finally, another current limitation is that . So far, we have methylation data from brain and blood, but not from other tissues. Therefore, we cannot determine whether the brain-specific DMRs are exclusive to the brain. It would be of great interest to collect other tissues and investigate secondary damage in cardiac arrest.

---

## [Decision Letter · Decision Letter 2]

10 Mar 2025

The application of cell-free DNA methylation patterns in critical illnesses: A protocol paper

PONE-D-24-42835R2

Dear Dr. David,

We’re pleased to inform you that your manuscript has been judged scientifically suitable for publication and will be formally accepted for publication once it meets all outstanding technical requirements.

Kind regards,

Elingarami Sauli, PhD

Academic Editor

PLOS ONE

Additional Editor Comments (optional):

The authors have satisfactorily addressed all major and minor comments by the reviewers. This submission can now be accepted after grammar/typo corrections by the authors.

Reviewers' comments:

Reviewer's Responses to Questions

**Comments to the Author**

1. Does the manuscript report a protocol which is of utility to the research community and adds value to the published literature?

Reviewer #3: Yes

2. Has the protocol been described in sufficient detail?

To answer this question, please click the link to protocols.io in the Materials and Methods section of the manuscript (if a link has been provided) or consult the step-by-step protocol in the Supporting Information files.

The step-by-step protocol should contain sufficient detail for another researcher to be able to reproduce all experiments and analyses.

Reviewer #3: Yes

3. Does the protocol describe a validated method?

Reviewer #3: Yes

4. If the manuscript contains new data, have the authors made this data fully available?

Reviewer #3: Yes

**5. Is the article presented in an intelligible fashion and written in standard English?**

Reviewer #3: Yes

6. Review Comments to the Author

Reviewer #3: The revised manuscript can be accepted for publication after minor corrections:

Could you please specify what Zyagen is? How much plasma was used for cell-free DNA extraction, all the 10 mL?

Please correct the typo in this sentence: "Finally, another current limitation is that . So far, we have methylation data from brain and blood, but not from other tissues."

7. PLOS authors have the option to publish the peer review history of their article (what does this mean? ). If published, this will include your full peer review and any attached files.

**Do you want your identity to be public for this peer review?** For information about this choice, including consent withdrawal, please see our Privacy Policy .

Reviewer #3: No

---

## [Editor Report · Acceptance letter]

PONE-D-24-42835R2

PLOS ONE

Dear Dr. Jang,

I'm pleased to inform you that your manuscript has been deemed suitable for publication in PLOS ONE. Congratulations! Your manuscript is now being handed over to our production team.

Kind regards,

on behalf of

Dr. Elingarami Sauli

Academic Editor

PLOS ONE